# Quantum LDPC Codes Based on Cocyclic Block Matrices

**DOI:** 10.3390/e25091309

**Published:** 2023-09-08

**Authors:** Yuan Li, Ying Guo

**Affiliations:** 1School of Electronic Information Engineering, Shanghai Dianji University, Shanghai 200240, China; 2School of Computer Science and Engineering, Beijing University of Posts and Telecommunications, Beijing 100876, China; guoying@bupt.edu.cn

**Keywords:** long-length quantum codes, cocyclic block matrices, stabilizer codes, low-density parity check codes

## Abstract

Motivated by a family of binary cocyclic block matrices over GF(2), we proposed a construction method to gain the stabilizer of long-length quantum error-correction codes (QECCs). Stabilizer quantum codes (SQCs) can be obtained by the different rows of the yielded circulant permutation matrices; hence, the quantum codes have the virtue of a fast construction algorithm. The recursive relation of a block matrix is employed in the proposed approach, so that the generator matrix of quantum cocyclic codes with long length can be constructed easily. Furthermore, the obtained quantum codes have the low-density advantage of there being no 4-cycles in the Tanner graph.

## 1. Introduction

The theoretical and experimental research on virtual quantum secure communication has gradually matured. However, due to the constraints of single photon sources, single photon detectors and the high cost of quantum devices, its practical application still faces many technical challenges. Extending the error correction coding technology in classical communication to the field of quantum information, the breakthrough in quantum error correction coding technology will undoubtedly become an important link in the practical application of quantum secure communication in the future, although quantum error-correcting codes (QECCs) are of only nominal significance at present in both quantum key distribution (QKD) and quantum secure direct communication (QSDC) [1,2]. Therefore, QECCs play theoretically a vital role in quantum information fields owing to their effects in fighting decoherence in quantum computing large-scale data [3,4,5,6,7,8,9,10]. The appeared errors from quantum channel noise and possible eavesdropper’s interference can be corrected by good QECCs. These codes are useful for building quantum fault-tolerant circuits. In general, an important class of quantum codes called stabilizer codes is firstly proposed by Gottesman [3]. Currently, in quantum coding theory, the stabilizer formalism of QECCs and its special form called Calderbank–Shor–Steane (CSS) are its main construction methods [4,5,6]. In quantum code construction theory, how to design the generator matrix resort to block matrix is the key. However, the error correction ability of short quantum codes is limited. The construction method of long-length quantum codes is becoming more important. In order to achieve high error correction ability, low-density parity check (LDPC) codes can achieve the maximum rate of reaching the Shannon limit with an error probability approaching zero when the codeword length tends to infinity. Since Gallager first proposed LDPC codes in the 1960s [11], two decades later, it caught the considerable interest of many researchers again [12,13,14,15]. Subsequently, with the rapid development of quantum information, many research achievements in its quantum versions have been made in this field [16,17,18,19,20,21,22,23]. However, compared to the in-depth study of classical LDPC codes, the corresponding quantum LDPC codes are still relatively less studied. One main obstacle is the difficulty of obtaining a more effective algorithm of iterative coding. With the application of large-scale data in quantum information fields, it becomes more important to encode massive data by making use of LDPC codes.

In quantum coding theory, the main problem for constructing low-density quantum code is how to obtain the generator for the stabilizer. Based on the current research results in this field, we will further study the encoding methods of quantum long-codes stabilizers. In this paper, a class of quantum stabilizer codes spirited with a family of cocyclic matrices is proposed [24,25]. The involved cocyclic block matrices are based on the Jacket matrix, of which the main property is that its inverse matrix can be gained by block-wise inverse [26,27]. According to the properties, its physical decoder circuit can be realized relatively simply. Meanwhile, the constructed quantum codes also have no 4-cycles in their Tanner graphs, so their parity check matrices have sparse characteristics.

The remainder of the paper is arranged as follows. In Section 2, we provide some necessary background and notations for QECCs. Then, in Section 3 and Section 4, in terms of circulant permutation matrices, we investigate the construction of quantum codes generated from cocyclic block matrices which are applied to construct a generator matrix of quantum codes. Furthermore, its poverties of low density are also analyzed in this section. Finally, some concluding remarks are drawn in Section 4.

## 2. Preliminaries

In this section, we briefly follow here the relevant terms and notation for quantum error-correcting codes and its stabilizer, which are used to be constructed in the paper.

### 2.1. General Construction Methods of QECCs

In terms of the depolarizing channel, there are four basic operators performing on a single qubit, i.e., $\sigma_{0}$, $\sigma_{x},\sigma_{z}$ and $\sigma_{y}=\sigma_{x}\sigma_{z}$, that can form a set P whose eigenvalues are ±1 and ±i; here, $\sigma_{x},\sigma_{z}$ are the Pauli transformations. The stabilizer formalism of quantum codes can provide a compact description of codewords, especially for larger codes. Generally, a stabilizer *S* of quantum code C[[N,k,d]] is a nontrivial subgroup of the *N*-operator group
(1)$$\begin{array}{c} P^{N}=P^{⊗N}=\left\{i^{\lambda }\left(\tau_{1}⊗\cdots ⊗\tau_{N}\right)\right|\tau_{j}\in E^{*},0\leq \lambda \leq 3,\\ 1\leq j\leq N\},\;\;\; \end{array}$$
here, any element in $E^{*}=\left\{\sigma_{0},\sigma_{x},\sigma_{z},\sigma_{y}\right\}$ is a Pauli matrix. It shows that the set $P^{N}$ consists of all tensor products of Pauli matrices on *N* qubits again with multiplicative factors, which is a Hilbert codeword space of *N* binary codes. Therefore, for a stabilizer code C of encoding *k* logical qubits into *N* physical qubits, there are $2^{N-k}$ independent elements in $2^{k}$-dimension code space C(S) associated with *S* of form
(2)$$\begin{array}{c} C(S)=\{|\psi \rangle :M|\psi \rangle =|\psi \rangle ,\forall M\in S\}. \end{array}$$To obtain the generator matrix of a stabilizer, two constraints should be met. One is that any pair of elements in stabilizer *S* commute, and the second condition implies that all elements of *S* are Hermitian, i.e., they have an eigenvalue of ‘±1’. It means that *S* is specified by N−k commuting and independent generators, $\left\{M_{i}:1\leq i\leq N-k\right\}$. Correspondingly, if stabilizer quantum code is to be formed, N−k generators of stabilizer *S* should be firstly gained, which are described by matrix
(3)$$\begin{array}{cc} G_{N}=[G^{x} & G^{z}{]}_{(N-k)\times 2N}={\left[M_{1},M_{2},\ldots ,M_{N-k}\right]}^{T}, \end{array}$$
where $G^{x}={\left(g_{ij}^{x}\right)}_{\left(N-k\right)\times N},\;\;G^{z}={\left(g_{ij}^{z}\right)}_{\left(N-k\right)\times N}$ for 1≤i≤N−k,1≤j≤N. According to the property of commuting generators, the elements of its row vector satisfy that the symplectic inner product is equal to zero. It means that an arbitrary operation $\gamma_{i}$ in the *N*-qubit depolarizing channel belongs to $G_{N}$, which may be uniquely shown by
(4)$$\begin{array}{c} \gamma_{i}=\tau \left[\sigma_{1}^{x_{i1}}\sigma_{1}^{z_{i1}}\right]⊗\left[\sigma_{2}^{x_{i2}}\sigma_{2}^{z_{i2}}\right]⊗\ldots ⊗\left[\sigma_{N}^{x_{iN}}\sigma_{N}^{z_{iN}}\right] \end{array}$$
where $x_{i_{s}},z_{i_{s}}\in \left\{0,1\right\}$ for 1≤s≤N, and τ is a factor among the eigenvalues. Without considering the factor τ, based on single-qibit operator $\gamma_{i}$, we denote a 2N-dimensional concatenated vector as
(5)$$\begin{array}{c} {\vec{\gamma }}_{i}=\left({\vec{x}}_{i}|{\vec{z}}_{i}\right)=\left(x_{i1},x_{i2},\ldots ,x_{iN}|z_{i1},z_{i2},\ldots ,z_{iN}\right). \end{array}$$For any two concatenated row vectors ${\vec{\gamma }}_{i}=\left(x_{i}|z_{i}\right)$ and ${\vec{\gamma }}_{v}=\left(x_{j}|z_{j}\right)$ in generator matrix $G_{N}$, the symplectic inner product is defined by
(6)$$\begin{array}{c} {\vec{\gamma }}_{i}\cdot {\vec{\gamma }}_{j}={\vec{x}}_{i}{\vec{z}}_{j}+{\vec{x}}_{j}{\vec{z}}_{i}=\sum_{k=1}^{N}\left(x_{ik}z_{jk}+x_{jk}z_{ik}\right). \end{array}$$Thereby, two operations ${\vec{\gamma }}_{i}$ and ${\vec{\gamma }}_{j}$ satisfy the commuting condition if
(7)$$\begin{array}{c} {\vec{\gamma }}_{i}\cdot {\vec{\gamma }}_{j}=0. \end{array}$$The vector ${\vec{\gamma }}_{i}$ in Equation (Equation 5) is generally used to seek the generators of the stabilizer.

### 2.2. Error of Quantum Error-Correction Code and Bound

All errors that occur in different quantum codes can be labeled with string vector over field $F_{2}$ when the states are transmitted through quantum channels. Single qubit error correction for quantum codes includes three classes, i.e, flip error, phase error or phase–flip error, which can be described with three corresponding Pauli operators: $\sigma_{x}$, $\sigma_{z}$ and $\sigma_{y}$, respectively. Assuming a single error qubit described as flip error X(a) and phase error Z(a) for $a\in F_{2}$, a state |x〉∈C acted by error is denoted as
(8)$$X\left(a\right)|x\rangle =|x+a\rangle \;\;\;\;\;\mathrm{and}\;\;\;Z\left(a\right)|x\rangle =\omega^{a*x}|x\rangle .$$Correspondingly, the errors on *N* qubits can be denoted as vector $e=\sigma_{x}^{X}\sigma_{z}^{Z}$, for $X=(x_{1},x_{2},\ldots ,x_{N})$ and $Z=\left(z_{1},z_{2},\ldots ,z_{N}\right)\in F_{2}^{N}$. Hence, it acts on an *N*-qubit basis state $|Q\rangle =\left(q_{1},q_{2},\ldots ,q_{N}\right)$ over $F_{2}^{N}$
(9)$$\begin{array}{c} e|Q\rangle ={\left(-1\right)}^{Z\cdot Q}|X+Q\rangle ={\left(-1\right)}^{\sum_{i=1}^{N}z_{i}\cdot q_{i}}|\sum_{i=1}^{N}x_{i}+q_{i}\rangle . \end{array}$$

Furthermore, the quantum Hamming bound of a general quantum code C[[N,k,d]] is required to satisfy the following inequality condition [28]
(10)∑l=0t3lNl≤2N−k,
which may correct up to t=[(d−1)/2] quantum error bits.

## 3. Quantum Codes Based on Cocyclic Quasi-Jacket Block Matrices

Motivated by the center weight Hadamard matrices [26], a family of orthogonal matrices called Jacket transform was proposed [27], which is widely employed in signal processing, encoding, mobile communication, etc. [29,30,31,32]. Especially, Hadamard and DFT matrices belong to this kind of Jacket matrix.

### 3.1. Definitions and Terms

Based on the general definition of a square matrix, in this section, we presented a construction method of quantum codes over binary matrices. As the following, for obtaining the stabilizer generators of quantum codes over the complex number field with a bigger size, we consider a class of binary cocyclic Jacket matrices based on a circular permutation matrix to construct its concatenated generator matrix, i.e.,
(11)$$\begin{array}{c} G_{N}={\left[G^{x}|G^{z}\right]}_{N\times 2N}, \end{array}$$
here, $G^{x}$ and $G^{z}$ are generator matrices of the encoding bit and phase, respectively.

Since a cocyclic approach was introduced in [24], some interesting codes come from useful cocyclic matrices, such as the Hadamard matrix that is the matrix representation of a cocycle [25].

**Definition** **1.**
*If set G is a finite group of order v and A is an Abelian group of order w, respectively, a mapping: φ: G × G →A is called a cocycle while it satisfies the following particular cocycle equation*

(12)
$$\begin{array}{c} \varphi (g,h)\varphi (gh,k)=\varphi (g,hk)\varphi (h,k),\;\;\;\forall g,h,k\in G. \end{array}$$



From the definition, it is easy to show that φ(g,1)=φ(1,h)=φ(1,1),∀g,h∈G, so φ(1,1)=1, as we call it, is a normalized cycle.

According to the cocycle mapping, if the rows and columns of a square matrix can be indexed by the elements of G in terms of some distribution, the matrix is called a cocyclic matrix, which is described as
(13)$$\begin{array}{c} M_{\varphi }={\left[\varphi \left(g,h\right)\right]}_{g,h\in G}, \end{array}$$
where the map φ(g,h) denotes a position (g,h) of the entry in matrix $M_{\varphi }$. If the cocycle φ is symmetric, $M_{\varphi }$ will be a symmetric matrix. Furthermore, on the basis of the cocylic matrices, some cocyclic codes are primitively derived from Hadamard matrices or from related block designs [33]. In this case, a v×v Hadamard matrix *H* with entries {±1} such that $HH^{T}=vI_{v}$ is used to construct codes with length v=4s for integer s≥1. If 1s and −1 s are replaced by 1 s and 0 s, respectively, the Hadamard matrix becomes a binary one. A main superiority of cocyclic codes is that the long-length codes may be easily gained with a circulant manner; hence, it may be applicable to the generalized classical case and further quantum codes.

**Definition** **2.**
*A square matrix $J_{N}={\left[a_{ij}\right]}_{N}$, if its inverse is obtained simply by element-wise inverse, i.e., it satisfies*

(14)
$$\begin{array}{c} a_{ij}^{-1}=\left\{\begin{array}{cc} a_{ij}/a & \mathrm{if}a_{\mathrm{ij}}\neq 0,\\ 0 & \;\;a_{ij}=0, \end{array}\right. \end{array}$$

*where a is the normalized constant, we call matrix $J_{N}$ a block Jacket matrix. Obviously, the Hadamard matrix belongs to the defined Jacket matrix. Furthermore, if there is a permutation matrix T, and its inverse matrix can be described as $J_{N}^{-1}=T\cdot {\left[a_{ij}\right]}_{N}/a$, matrix $J_{N}$ is a quasi-Jacket block matrix [34]. This class of matrices may be generalized by a simple binary transformation. Matrix decomposition with a Kronecker product of identity matrices can successively design a large length of codes that resorts to lower-order coefficient matrices. Meanwhile, this factorization decomposition generates a family of block circular sparse Jacket matrices which have good performance of LDPC codes. In the applications, the derived circulant permutation matrices with a Jacket pattern could lead to a very simple encoding algorithm.*


### 3.2. Cocyclic Matrix Derived from Jacket Block Matrices

On the basis of the above definitions, we shall apply the index mapping to construct generator matrices $G^{x}$ and $G^{z}$ based on cocyclic Jacket matrices with order $N=p^{m}$. Firstly, we form an Abelian group by factoring a quasi-Jacket block matrix, of which elements will be used to construct a stabilizer.

**Theorem** **1.**
*Given $A_{u}={\left[a_{ij}\right]}_{u}$ and $B_{v}={\left[b_{ij}\right]}_{v}$ are two cocyclic Jacket matrices, their Kronecker product $A_{u}⊗B_{v}$ will be also a cocyclic Jacket matrix with size uv [35].*


**Proof.** Because $A_{u}$ and $B_{v}$ are Jacket matrices, in terms of the definition of Jacket matrix, it has
(15)$$\begin{array}{c} A_{u}^{-1}={\left[a_{ij}^{-1}\right]}_{u}^{T}/c_{a},\;\;B_{v}^{-1}={\left[b_{sl}^{-1}\right]}_{v}^{T}/c_{b}, \end{array}$$
where $c_{a},c_{b}$ are two constants. As a result, their Kronecker product is $A_{u}⊗B_{v}={\left[w_{iu+s,ju+l}\right]}_{uv}$ for $w_{iu+s,ju+l}=a_{ij}b_{st}$. Correspondingly, we have
(16)$$\begin{array}{c} {\left(A_{u}⊗B_{v}\right)}^{-1}={\left[{\left(a_{ij}b_{st}\right)}^{-1}\right]}_{uv}^{T}/\left(c_{a}c_{b}\right)\\ ={\left[{\left(w_{iu+s,ju+l}\right)}^{-1}\right]}_{uv}/\left(c_{a}c_{b}\right), \end{array}$$
i.e., $A_{u}⊗B_{v}$ is a Jacket matrix.On the other hand, suppose that the row and column index orders of the two matrices as
(17)$$\begin{array}{c} g_{s1}^{a}\to g_{s2}^{a}\to \ldots \to g_{su}^{a},\;\;\;for\;\;g_{sj}^{a}\in G_{A}\\ g_{s1}^{b}\to g_{s2}^{b}\to \ldots \to g_{su}^{b},\;\;\;for\;\;g_{sl}^{b}\in G_{B} \end{array}$$
where s=r,c, and $g_{s1}^{a}$ (or $g_{s1}^{b}$) and $g_{c1}^{\tau }$ (or $g_{c1}^{b}$) refer to the first row and column indices of matrix $A_{u}$ (or $B_{v}$), respectively. Hence, the row and column index orders of matrix $A_{u}⊗B_{v}$ are
(18)$$g_{sj}^{a}g_{sl}^{b}\mapsto g_{si}^{a}g_{sh}^{b},\;\;\mathrm{if}\;\;g_{sj}^{a}\to g_{si}^{a}\;\mathrm{or}\;g_{sj}^{a}=g_{si}^{a},\;\;g_{sl}^{b}\to g_{sh}^{b}.$$Correspondingly, define
(19)$$\varphi_{AB}\left(g_{ri}^{a}g_{rh}^{b},g_{cj}^{a}g_{cl}^{b}\right)=\varphi_{A}\left(g_{ri}^{a}g_{cj}^{a}\right)\varphi_{B}\left(g_{rh}^{b}g_{cl}^{b}\right).$$Because both $A_{u}$ and $B_{v}$ are cocyclic matrices, $\varphi_{A}$ and $\varphi_{B}$ satisfy the condition shown in Equation (Equation 12). As a result, it is verified that $\varphi_{AB}$ also meets it with the index orders shown in Equation (Equation 18). It means that $A_{u}⊗B_{v}$ also is a cocyclic matrix. □

More generally, by making use of the mathematical methods of induction, it is easily verified that a large-length matrix
(20)$$J_{N}=J_{p_{1}}⊗J_{p_{2}}⊗\cdots ⊗J_{p_{n}}$$
with order $N=p_{1}p_{2}\cdots p_{n}$ is still a Jacket matrix and cocylic matrix, while Jacket matrix $J_{p_{i}}$ is a cocylic matrix for 1≤i≤n. Specially, if $p_{1}=p_{2}=\cdots =p_{N}$, the following matrix
(21)$$J_{p^{m}}≜\left[\omega^{\vec{i}\circ \vec{j}}\right]=J_{p}^{⊗n}$$
with a Kronecker product also achieves the definitions of Jacket matrix and cocylic matrix. In the following, we further illustrate that the generator matrix may be derived from a recursive cocyclic matrix.

**Theorem** **2.**
*A recursive p-order cocyclic matrix $J_{p^{m}}$ for a integer m can be composed of a family of cocyclic matrices $A=\{A_{N}^{1},A_{N}^{2},\cdots ,A_{N}^{m}\}$ with conventional multiplication as follows*

(22)
$$\begin{array}{c} J_{p^{m}}=\prod_{i=1}^{m}A_{p^{m}}^{i}=A_{p^{m}}^{1}A_{p^{m}}^{2}\cdots A_{p^{m}}^{m}, \end{array}$$

*where*

(23)
$$\begin{array}{c} A_{p^{m}}^{i}=I_{p^{i-1}}⊗J_{p}⊗I_{p^{n-i}}, \end{array}$$

*for that, $J_{p}$ is a cocyclic matrix and $I_{p}$ is an identity matrix.*


**Proof.** We use the mathematical induction method to prove the result. In fact, while n=1, the conclusion is obviously valid, i.e., $J_{p}=I_{p^{0}}⊗J_{p}⊗I_{p^{0}}$. Since the hypothesis is true for integer n=m, we consider that it holds for n=m+1. Namely, the recursive equation
(24)$$\begin{array}{c} J_{p^{m+1}}=J_{p^{m}}⊗J_{p}=\left(\prod_{i=1}^{m}A_{p^{m}}^{i}\right)⊗J_{p} \end{array}$$
holds. According to the property of the Kronecker product, it obeys the following formula
(25)(A⊗B)(C⊗D)=(AC)⊗(BD)
for any square matrices A,B,C and *D*. Hence,
(26)$$\begin{array}{c} \prod_{i=1}^{m+1}A_{p^{m}}^{i}=\prod_{i=1}^{m}\left(A_{p^{m}}^{i}⊗I_{p}\right)\left(I_{p^{m}}⊗J_{p}\right)=\prod_{i=1}^{m}A_{p^{m}}^{i}⊗J_{p}. \end{array}$$It means that Equation (Equation 24) is equal to Equation (Equation 26), i.e., the case of n=m+1 also holds. □

In terms of the above conclusion, based on its decomposition factors, we can form a *factor group* $A=\{A_{N}^{1},A_{N}^{2},\cdots ,A_{N}^{m}\}$. Resorting to the factor group A, we consider constructing the stabilizer with two isomorphic classes of groups.

**Lemma** **1.**
*Take factor group A in Equation (Equation 23) and its isomorphic factor group $B=\{B_{N}^{1},B_{N}^{2},\cdots ,B_{N}^{m}\}$ with matrix $J_{p}^{\prime }$. In terms of the property of the Kronecker product, if matrices $J_{p}$ and $J_{p}^{\prime }$ satisfy the commuting condition, i.e., their symplectic inner product is equal to zero, it is easy to check that any pair of factors $A_{N}^{i}$ and $B_{N}^{i}$ (1≤i≤m) in the two groups also satisfies the commuting condition (Equation 7).*


It means that any pair of $A_{N}^{i}$ and $B_{N}^{i}$ can be taken as the generator matrices $G^{x}$ and $G^{z}$ of quantum codes, respectively, when two matrices $J_{p}$ and $J_{p}^{\prime }$ commute.

**Example** **1.**
*If the prime integer is given p=2, let $\omega =e^{\sqrt{-1}\pi /2}$. It has vectors ${\vec{i}}_{0}=\left(1=\omega^{0},\omega \right)$ and ${\vec{i}}_{1}=\left(\omega ,1\right)$ in group G={1,ω}.*


#### 3.2.1. Quasi-Jacket Block Matrices with Size $4^{m}$ by $4^{m}$

Resorting to the binary matrixes, we consider building the generator matrix of quantum stabilizer codes. On the one hand, we define a class of fundamental matrices α and β derived from binary Hadamard matrix *H* as follows, respectively:(27)$$\begin{array}{ccc} & & \alpha ≜\left[\begin{array}{cc} 1 & 0\\ 1 & 1 \end{array}\right],\;\;\;\beta ≜\left[\begin{array}{cc} 1 & 1\\ 0 & 1 \end{array}\right]. \end{array}$$It is easy to check that the introduced matrices have the following properties
(28)$$\begin{array}{c} \alpha \beta +\beta \alpha =I_{2},\;\;\;\alpha \beta \times \beta \alpha =I_{2}.\;\;\alpha^{2}=\beta^{2}=I_{2}. \end{array}$$Furthermore, define a permutation matrix with 2q order as
(29)$$\begin{array}{c} T_{2q}=\left[\begin{array}{cc} 0 & 1\\ 1 & 0 \end{array}\right]\times I_{q}. \end{array}$$

Based on the above fact, with simple computation, we can obtain a 4-order matrix and its inverse matrix
(30)$$\begin{array}{c} J_{4}≜\left[\begin{array}{cc} \alpha & \beta \\ \beta & \alpha \end{array}\right]\;\mathrm{and}\;\;J_{4}^{-1}=\left[\begin{array}{cc} \beta & \alpha \\ \alpha & \beta \end{array}\right]=T_{4}\times J_{4}, \end{array}$$
which is a quasi-Jacket block matrix in terms of the definition.

Similarly, we may also obtain the 16-order quasi-Jacket block matrix as
(31)$$\begin{array}{c} J_{16}=J_{4^{2}}=J_{4}⊗J_{4}={\left[\begin{array}{cc} \alpha & \beta \\ \beta & \alpha \end{array}\right]}^{⊗2}\\ =\left[\begin{array}{cccc} 1 & 0 & 1 & 1\\ 1 & 1 & 0 & 1\\ ine1 & 1 & 1 & 0\\ 0 & 1 & 1 & 1 \end{array}\right]⊗J_{4}=\left[\begin{array}{cccc} J_{4} & O_{4} & J_{4} & J_{4}\\ J_{4} & J_{4} & O_{4} & J_{4}\\ ineJ_{4} & J_{4} & J_{4} & O_{4}\\ O_{4} & J_{4} & J_{4} & J_{4} \end{array}\right], \end{array}$$
and its inverse matrix
(32)$$\begin{array}{c} J_{16}^{-1}=T_{16}\times \left[\begin{array}{cccc} J_{4}^{-1} & O_{4} & J_{4}^{-1} & J_{4}^{-1}\\ J_{4}^{-1} & J_{4}^{-1} & O_{4} & J_{4}^{-1}\\ J_{4}^{-1} & J_{4}^{-1} & J_{4}^{-1} & O_{4}\\ O_{4} & J_{4}^{-1} & J_{4}^{-1} & J_{4}^{-1} \end{array}\right]. \end{array}$$According to its definition, the matrix obviously is a quasi-Jacket block matrix.

Generally, by resorting to the recursive relationships based on $J_{4}$ defined by (Equation 30), we can obtain the quasi-Jacket block matrices with larger size $4^{m}$ by using the Kronecker product, i.e.,
(33)$$\begin{array}{c} J_{4^{m}}=J_{4}⊗J_{4^{m-1}},\;\;\;\;\mathrm{for}\;\;m\geq 2, \end{array}$$
which may be used to generate the generator matrices $G^{x}$ (or $G^{z}$).

On the other hand, to obtain the matrix $G^{z}$ (or $G^{x}$), we consider another class of matrix called the permutation matrix $P^{h}={\left(p_{ij}\right)}_{p\times p}$,
(34)$$\begin{array}{c} p_{ij}=\left\{\begin{array}{cc} 1 & \mathrm{if}\;i=(j+h)\;\mathrm{mod}\;p,1\leq h<p,\\ 0 & \;\;\mathrm{otherwise}, \end{array}\right. \end{array}$$
for an even number p≥4. According to the definition, it is obviously a finite cyclic group $P=\{I,P,\ldots ,P^{p-1}\}$ formed by a family of matrices $P^{i},0\leq i\leq p-1$ is an Abelian group based on the matrix-multiplication operation, and $I=P^{0}$ is its unit element. Here, given that the number *p* is an even number and *h* is an odd number, respectively, a sequence of matrices whose exponents are odd can form a subset $P_{o}=\left\{P,P^{3},\ldots ,P^{p-1}\right\}$ of Abelian group P.

Based on the generated two classes of matrices, i.e., the cocyclic Jacket matrix and permutation matrix, we consider obtaining the stabilizer for quantum codes. Let $G^{x}$ with a size of $N=4^{m}$ be obtained by the cocyclic Jacket matrix described before. In the following, we consider how to construct another matrix $G^{z}$. Firstly, we take two 4-order basic permutation matrices $P_{4}$ in Equation (Equation 34) with parameters h=1,3 respectively, i.e., $\left\{\omega^{2},\omega^{4}\right\}$, such that
(35)$$\omega^{2}=P=\left[\begin{array}{cccc} 0 & 1 & 0 & 0\\ 0 & 0 & 1 & 0\\ 0 & 0 & 0 & 1\\ 1 & 0 & 0 & 0 \end{array}\right]\;\mathrm{or}\;\omega^{4}=P=\left[\begin{array}{cccc} 0 & 0 & 0 & 1\\ 1 & 0 & 0 & 0\\ 0 & 1 & 0 & 0\\ 0 & 0 & 1 & 0 \end{array}\right].$$Then, it provides that $P_{N}=P_{4}⊗P_{N-1}$, and it takes $G_{z}=P_{4}⊗P_{N/4}$. As a result, the concatenated generator matrix *G* is built.

In the following, we use the mathematical induction to deduce that two matrices $G^{x}$ and $G^{z}$ in the obtained concatenated generator matrix *G* satisfy self-orthogonal and commuting conditions, i.e., the obtained generator matrix should meet the conditions of self-orthogonal and commuting, so that it may be used to construct a stabilizer of quantum codes. In fact, it is obvious that matrix $G^{z}$ satisfies the self-orthogonal, so we firstly consider that the case of m=1 for $G^{x}$ meets the self-orthogonal constrain. Namely, in the following matrix
(36)$${\left[G^{x}|G^{z}\right]}_{4\times 8}=\left[J_{4}|P_{4}\right]=\left[\begin{array}{cccccccc} 1 & 0 & 1 & 1 & 0 & 0 & 0 & 1\\ 1 & 1 & 0 & 1 & 1 & 0 & 0 & 0\\ ine1 & 1 & 1 & 0 & 0 & 1 & 0 & 0\\ 0 & 1 & 1 & 1 & 0 & 0 & 1 & 0 \end{array}\right],$$
any two rows in $G^{x}$ and $G^{z}$ are self-orthogonal and commuting. Assuming it meets the case of m−1, we consider the result for *m*. In fact, if any two rows in concatenated block submatrix [αβ] meet the conditions, it can also do so in [βα]. Without a loss of generality, here, we only consider the rows in submatrix [αβ]. By making use of the Kronecker product of any two row vectors in $G_{4}^{x}$, it is easy to check that they meet the orthogonal condition.

And then, the symplectic inner product in Equation (Equation 6) is proved as following. Define complement operators $\bar{1}=0$ and $\bar{0}=1$, according to number theory; then, it is possible to easily check that matrices $\bar{P}=\left({\bar{p}}_{ij}\right)$ with parameters h=1,3 commute. Namely, any two row vectors in concatenated matrices $\left(\bar{P}|P\right)$ satisfy the commuting condition in Equation (Equation 7). In fact, it can be checked that the elements in the Abelian group P commute when their cycle indexes both are odd or even numbers. The matrices satisfy isomorphic J≅P when its corresponding orthogonal matrix also satisfies isomorphic $J≅\bar{P}$. The complement of $G^{x}$ is an element in the circulant permutation group $P_{0}$; hence, two pairs of vectors ${\vec{x}}_{1}$, ${\vec{x}}_{2}$ and ${\vec{z}}_{1}$, ${\vec{z}}_{2}$ all belong to this group. The product of any two vectors in $G^{x}$ and $G^{z}$ is equal to 0 or 1 simultaneously; i.e., it means that two pairs of vectors of the inner product with module 2 is equal to 0.

Similarly, we consider the generator matrix based on the block matrix $J_{16}$ and 16-order permutation matrix stemmed from $P_{4}$.
(37)$$\begin{array}{c} {\left[G^{x}|G^{z}\right]}_{16\times 32}=\left[J_{16}|P_{4}^{⊗2}\right]\\ =\left[\begin{array}{cccccccc} J_{4} & O_{4} & J_{4} & J_{4} & O_{4} & O_{4} & O_{4} & P_{4}\\ J_{4} & J_{4} & O_{4} & J_{4} & P_{4} & O_{4} & O_{4} & O_{4}\\ J_{4} & J_{4} & J_{4} & O_{4} & O_{4} & P_{4} & O_{4} & O_{4}\\ O_{4} & J_{4} & J_{4} & J_{4} & O_{4} & O_{4} & P_{4} & O_{4} \end{array}\right]. \end{array}$$

Generally, the *N*-order generator matrix for $N=4^{m},m\geq 2$ can be generated by
(38)$$\begin{array}{c} {\left[G^{x}|G^{z}\right]}_{N\times 2N}=\left[J_{N}|P_{4}⊗P_{N/4}\right]. \end{array}$$

On the other hand, from the factorization of a large number, $4^{m}$, in Equation (Equation 22), we can obtain a group A with Formula (Equation 23) whose element $A_{p^{m}}^{i}=I_{4^{i-1}}⊗J_{4}⊗I_{4^{n-i}}$. Any element $A_{4^{m}}^{i}$ in the group can be taken as the generator $G^{x}$. Similarly, we can form another group B based on $P_{4}$, and take the element $B_{4^{m}}^{i}$ (same structure to $A_{4^{m}}^{i}$) as the generator $G^{z}$. As a result, a large-length generator matrix with size $4^{m}$ by $4^{m}$ is gained.

#### 3.2.2. Quasi-Jacket Block Matrices with Size $4p^{m}$ by $4p^{m}$

Furthermore, to obtain the more general length of code, we define a matrix
(39)$$\begin{array}{c} Q_{\alpha }≜\left[\begin{array}{cc} I & O\\ P^{h} & I \end{array}\right],\;\;\;\;\;Q_{\beta }≜\left[\begin{array}{cc} I & P^{-h}\\ O & I \end{array}\right]. \end{array}$$It is easily checked that
(40)$$\begin{array}{c} Q_{\alpha }^{2}=Q_{\beta }^{2}=I_{2p},\;\;\;\;\mathrm{and}\;\;\;\;\;Q_{\alpha }Q_{\beta }+Q_{\beta }Q_{\alpha }=I_{2p}. \end{array}$$Furthermore, in terms of the basic properties, we consider matrix
(41)$$\begin{array}{c} Q≜\left[\begin{array}{cc} Q_{\alpha } & Q_{\beta }\\ Q_{\beta } & Q_{\alpha } \end{array}\right]=\left[\begin{array}{cccc} I & O & I & P_{p}^{-h}\\ P_{p}^{h} & I & O & I\\ I & P_{p}^{-h} & I & O\\ O & I & P_{p}^{h} & I \end{array}\right], \end{array}$$
and its inverse matrix
(42)$$\begin{array}{c} Q^{-1}=\left[\begin{array}{cc} Q_{\beta } & Q_{\alpha }\\ Q_{\alpha } & Q_{\beta } \end{array}\right]=\left[\begin{array}{cccc} I & P_{p}^{-h} & I & O\\ O & I & P_{p}^{h} & I\\ I & O & I & P_{p}^{-h}\\ P_{p}^{h} & I & O & I \end{array}\right]\\ =\left[\begin{array}{cc} O & I\\ I & O \end{array}\right]\times \left[\begin{array}{cc} Q_{\alpha } & Q_{\beta }\\ Q_{\beta } & Q_{\alpha } \end{array}\right]=T_{4p}\times Q. \end{array}$$It is obvious that the constructed matrix Q is also a quasi-Jacket matrix which can be seen as a 4p×4p binary cocyclic matrix. It means that a quantum code with a length of 4p may be obtained. Similarly, the recursive relationships may be used to obtain the large-length block matrices with a larger size $4p^{m}$ by using the Kronecker product, i.e.,
(43)$$\begin{array}{c} J_{4p^{m}}=J_{4}⊗J_{p^{m}},\;\;\;\;\mathrm{for}\;\;m\geq 2, \end{array}$$
such that the more general case of the generator matrix with a long size is achieved. In a similar manner, the obtained large-size matrix also satisfies the conditions of self-orthogonal and commuting like the process before proving the case of size $4^{m}$. We take two matrices based on parameters $h_{1},h_{2}$ in $P_{e}=\left\{\omega ,\omega^{3},\ldots ,\omega^{p-2}\right\}$ or in $P_{o}=\left\{\omega^{2},\omega^{4},\ldots ,\omega^{p-1}\right\}$ as $G^{x}$ and $G^{z}$, respectively. It is evident that the obtained matrices commute each other.

### 3.3. Quantum LDPC Codes Based on Cocyclic Block Matrix

To encode *k* information qubits into a QC quantum code C with parameter [[N,k]], one should firstly gain two N×N cocyclic circulant permutation matrices *G* based on the previously constructed matrices $G^{x}$ and $G^{z}$ of satisfying the orthogonal condition. According to the recursive relationship, we take any N−k rows of the two generated matrices as the generators $G^{x}$ and $G^{z}$. It means that the concatenated matrix $G_{N}$ in Equation (Equation 3) meets the construction condition of quantum code.

In classic coding theory, a class of very important linear codes called LDPC code is widely explored and used in practice. Its theoretical importance mainly embodies that its good performance of linear-time decoding can achieve Shannon capacity [14,16]. Its quantum version, known as quantum LDPC (QLDPC) codes, plays a very important role in quantum information. However, compared to its classical counterparts, the achieved fruits of quantum LDPC codes are still far less. The main problem is the lack of efficient methods for obtaining iterative coding. Generally, the proportion of 1s in a binary matrix is called the matrix’s destiny. Given that $\iota_{0}$ and $\iota_{1}$ are the numbers of 0s and 1s in a matrix, respectively, the destiny is
(44)$$\begin{array}{c} d_{0}=\frac{\iota_{1}}{\iota_{0}+\iota_{1}}. \end{array}$$According to the construction method described in Equation (Equation 41), it obvious that the total number of 0s and 1s in a 4p×4p matrix is $16p^{2}$. Furthermore, each of the 12 matrices $I,P^{-h}$ and $P^{h}$ in G contain *p* 1s; hence, there are 12p 1s in this matrix: i.e, the 1s density in G is
(45)$$\begin{array}{c} d_{4p}=\frac{12p}{16p^{2}}=\frac{3}{4p}. \end{array}$$From the ratio, it means that the constructed cocylic matrix has a good performance of low-density property. On the other hand, there are N(log2N−1) additions and 2N(log2N−1) multiplications for the computation complexity in processing the obtained matrix with size $N=4^{m}$. The check matrix *H* of LDPC codes is generally represented by a simpler and more intuitive Tanner graph with a one-to-one map. If there are equal degrees of all variable nodes in the Tanner graph, its corresponding LDPC code is called a regular code. Otherwise, it is called an irregular code. Denote by η and ρ the variable and check the nodes’ degrees corresponding respectively to the minimum weights of the row and column of the check matrix. Assume that the girth *g* in the graph refers to the length of the shortest cycle [36,37]. Then, the following bounds should be satisfied, i.e.,
(46)$$\begin{array}{c} N\geq 1+\sum_{i=2}^{x+1}\eta {\left(\eta -1\right)}^{i-2}{\left(\rho -1\right)}^{i-1} \end{array}$$
for girth g=4x+2, and
(47)$$\begin{array}{c} N\geq 1+\sum_{i=1}^{x+1}\eta {\left[\left(\eta -1\right)\left(\rho -1\right)\right]}^{i-1} \end{array}$$
for girth g=4x, where *x* is an integer, it can be seen from the above bounds that the code length will increase exponentially, while the cycle length and weights also are rising.

To obtain the good performance, having no 4-cycles in the Tanner graph of the quantum LDPC codes is required. A cycle is composed of a group of interconnected vertices in a Tanner graph, with one of these vertices serving as both the starting and ending points, and passing through each vertex only once. If the length of a cycle is called the number of lines it contains, the girth of a graph is defined as the minimum cycle length in the graph. The traditional cycle detection methods are generally based on block matrix sequences [15]. When using this algebraic description to detect cycles in a given check matrix, there is redundancy due to the possibility that different block matrix sequences may form the same girth. To avoid redundancy caused by repeated traversal, we propose the following method for detection.

**Theorem** **3.**
*Based on the cocyclic block matrices in Equations (Equation 38) and (Equation 43), the constructed quantum codes have no 4-cycles in the Tanner graph.*


**Proof.** According to the proposed construction method, matrix $J_{4}$ derived from matrices α and β obviously cannot form a cycle. If it considers a quasi-Jacket block matrix sequence with a cycle of girth 2i, $\left(j_{0},l_{0}\right);\left(j_{1},l_{1}\right);\ldots ;\left(j_{i-1},l_{i-1}\right);\left(j_{0},l_{0}\right)$ as the loop is cyclic. Here, elements $\left(j_{v},l_{v}\right)$ represent the block matrix of row $j_{v}$ and column $l_{v}$ for 0≤v≤i. The formation of a cycle means that any block matrix sequence $(j_{0},l_{i-1)};\left(j_{i-1},l_{i-2}\right);\left(j_{i-2},l_{i-3}\right);\left(j_{v+1},l_{v}\right)\left(j_{1},l_{0}\right)$ is different from the original sequence.Assume new coordinates $\left(J_{v},L_{v}\right)$ to describe any block matrix in a block matrix sequence. For a cycle with a length of 2i, the block matrices in the sequence can be located at most in the *i*-th row. Without losing generality, the sequence of block matrices that may generate a 2i girth is classified according to the number of possible rows. Assume that *i* block matrices are located in the u(2≤u≤i) row, and *u* rows are marked in order from top to bottom (1,2,…,u). The new row coordinate ranges $J_{v}\in \left\{1,2,\ldots ,u\right\}$ for each block matrix, as adjacent block matrices cannot be in the same row. Therefore, we can obtain $J_{v}\in \left\{1,2,\ldots ,m\right\}\backslash J_{v-1}$, where 2≤v<i. When v=i, the *i*-th block matrix is adjacent not only to the *i*-th block matrix but also to the first block matrix, so $J_{i}\in \left\{1,2,\ldots ,u\right\}\backslash \left\{J_{1},J_{i-1}\right\}$ can be obtained. For column coordinates, making $L_{1}=1$, “1” does not mean that the block matrix is in the first column, but rather that the column in which the block matrix is located is the first occurrence in the sequence. The subsequent block matrix is marked with new column coordinates $L_{v}$ in the order. We implement this by setting the parameter $M_{v}=\max_{i=1:v}L_{i}$, where $M_{v}$ represents the number of columns distributed in the first *v* block matrices. Then, for the *v*-th block matrix, $L_{v}\in \left\{1,2,\ldots ,M_{v-1},M_{v-1}+1\right\}\backslash L_{v-1}$ will indicate that the column can be in a different column from the first v−1 block matrix. For the i-th block matrix located in row u(2≤u≤i), the first block matrix will be $J_{1}=L_{1}=1$, i.e., the new coordinate is (1,1) and $M_{1}=1$. For the *v*-th block matrix, we may obtain that $L_{v}\in \left\{1,2,\ldots ,M_{v-1},M_{v-1}+1\right\}\backslash L_{v-1}$; hence, the new coordinates of the current block matrix sequence have (u−1)×(v−1) possibilities. After traversing each possibility and determining all the branches of the *i*-th block matrix, we can obtain that the cycle of the constructed matrix is bigger than 4 when the parameter in construction method meets m≥2.According to the above description, after determining all the branches of the *i*-th block matrix, a tree diagram of the i-th layer will be obtained, thereby avoiding redundancy caused by repeated traversal. □

In the classical coding field, we firstly compare the performance of the classical codes obtained from the cocyclic block matrices with the classical quasi-cyclic (QC) codes in Figure 1. In light of the character of the quasi-Jacket block matrix, the gained generator matrix of classical code has sparse property; hence, the parity check matrix of low weight can be obtained. In this figure, we take the different cocyclic matrices presented in this paper to analyze their bit error ratio (BER) and consider the coding performance of a single one among the generated pair classical codes. Here, given the different prime numbers *p* are taken n=5,7 with the construction method of the quasi-Jacket block matrix in this paper, their corresponding quantum code lengths are $N=4p^{m}=500,1372$ for m=3, respectively. Assuming the binary phase shift keying (BPSK) modulation is over an additive white Gaussion noise (AWGN) channel, the bit error performance of the selected quantum codes is shown in this figure. As a result, the gained codes have the good properties of a sparse matrix. In decoding the performance of LDPC, short cycles will degrade the result under an iterative algorithm. The fewer short cycles of code makes its performance better; hence, the prime numbers p=5,7 more than 4 are taken to avoid 4-cycles in the Tanner graph.

It can be seen from the result that the constructed quantum LDPC codes have no apparent performance in the low signal-to-noise ratio (SNR) region, but the cocyclic codes are better than quasi-cyclic codes at high SNR. Furthermore, we also can see that the difference becomes more pronounced when the code length is larger. Therefore, the designed cocyclic codes may be better applied to construct large-length quantum codes.

Generally, three types of independent Pauli errors occur in a realistic depolarizing communication channel, i.e., bit-flip errors *X*, phase–flip errors *Z* and *Y*, and errors that are the combination of *X* and *Z*. Assuming that the total probability denotes as *f*, the probabilities of the three classes of errors are equally f/3 owing to their independence. Namely, the distributed marginal flip probability can be depicted as $f_{m}=2f/3$. As a result, the errors of types *X* and *Z* in the obtained generator matrices $G^{x}$ and $G^{z}$ are separately corrected by reason of their isomorphism.

Furthermore, in the construction of quantum LDPC codes, the binary generator matrix of a quantum code is constructed by matrices $H^{x}$ and $H^{z}$ which can be seen as the parity-check matrices of two two classical codes $C_{x}$ and $C_{z}$, respectively. The spaces of degenerate *X*-codewords $C_{x}^{⊥}$ and *Z*-codewords $C_{z}^{⊥}$ are derived from the rows of matrices $H^{x}$ and $H^{z}$. The minimum distance $d=\min\{d_{x},d_{z}\}$ of the quantum code is defined as the smaller of two-types distances $d_{x}$ and $d_{z}$. According to the structure of the generator matrices, at least the minimum distances are bounded above by $O(N^{1/2}\log N)$ as the code length N→∞.

In the following, we compare the proposed quantum cocyclic codes to the quantum Reed-Solomon coding method with the previous work in ref. [38] under the 0.5 coding rate of constructed codes in Figure 2. The designed quantum codes with lengths N=500,1372, respectively, and similar length of CSS-type quantum Reed-Solomon (RS) codes are taken for comparison. Here, we take the generator in terms of the similar method from Equation (Equation 31) with size N=1024,4096 for parameter m=5,6 respectively. Furthermore, similar lengths of quantum Reed–Solomon codes are constructed according to the circulant permutation of two classical codes in its construction method. Here, we take the prime number p=31,61, so that the codes of size N=930,3660, respectively, are gained for comparison.

In the SNR range, it is shown that the taken two shorter lengths of quantum codes have better performance from 0.2 to 0.4 dB than the longer one. However, owing to the close cycles and their bit-flip error correction capability, the higher length of code shows its superiority with increasing length.

## 4. Conclusions

The coding method of massive data has important applications in the field of information transmission and big-data processing. With the rapid development of quantum computing, this class of coding construction becomes one of the research fields of quantum information. In this paper, based on a class of circulant permutation matrices, we presented quantum cocyclic stabilizer codes in terms of two long-length block matrices. Motivated by the advantage of convenient and low-complexity implementation, a special family of Jacket block matrices is applied to the generator construction of a stabilizer. The presented method can construct long-length quantum cocyclic codes based on the iterative coding process. If the involved parameters are designed appropriately, the gained quantum codes have no 4-cycles in their generators, and the LDPC codes have good performance. In terms of the analysis for marginal flip probability $f_{m}$ about our proposed codes, the good performance of the sparse matrices that appeared with a bit error rate is shown in Section IV. Therefore, the obtained quantum cocyclic stabilizer codes can be potentially applied to the future quantum information field of processing massive data.

## Figures and Tables

**Figure 1 entropy-25-01309-f001:**
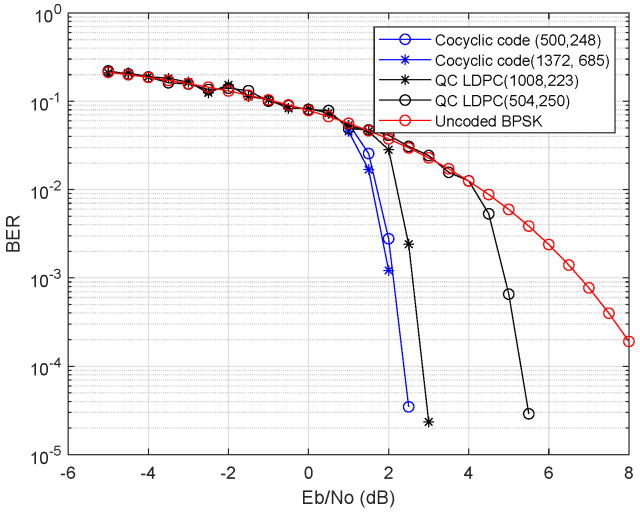
BER performance comparison of classical cocyclic code with N=500,1372 and classical quasi-cyclic codes with code rate R=0.5. To achieve the bigger length of cycle, the prime numbers p=5,7 more than 4 (for avoiding 4-cycles) are taken in the construction method. If the input is taken in the model of the Gaussian white noise channel (GWNC), we apply the quasi-Jacket block matrix to construct generator matrices. On the other hand, the quasi-cyclic codes with similar length are also constructed.

**Figure 2 entropy-25-01309-f002:**
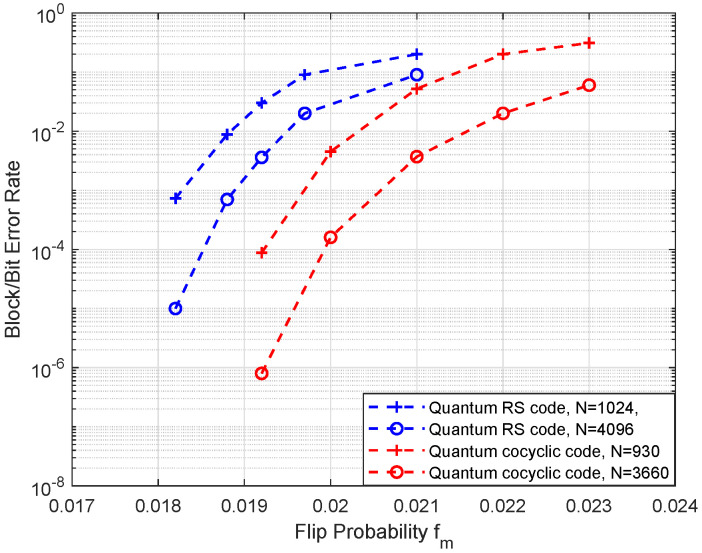
Comparison of decoding error rate of the proposed quantum cocyclic codes (red) and conventional quantum Reed–Solomon LDPC codes (blue). Here, the proposed quantum codes with lengths N=930,3660 are generated by m=5,6 on a binary symmetric channel, and the conventional quantum LDPC codes are obtained with similar lengths with rates $R_{q}$ = 0.50, respectively. For simplicity, assuming the equal error rate of two kinds of errors *X* and *Z*, only the bit error rate of flip probability $f_{m}$ is shown here.

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
