# Peer review of "Quantum LDPC Codes Based on Cocyclic Block Matrices"

_entropy, 2023, doi:10.3390/e25091309_

Round 1
Reviewer 1 Report
In this work, the authors proposed a construction method to get the stabilizer of long-length quantum error-correction codes (QECC). The stabilizer quantum codes (SQCs) can be obtained by the different rows of the yielded circulant permutation matrices, which makes the quantum codes good at realizing fast construction algorithm. They used recursive relation of block matrix so that the generator matrix of quantum cocyclic codes with long length can be constructed. The obtained quantum codes has low-density advantages of no 4-cycles in the Tanner graph. This is an interesting result, and can be used in quantum communication and computing. I recommend its acceptance after revision.
During the revision, the English should be improved. It should be pointed that QEEC is of only nomial siginificance at present because in both QKD and QSDC, no quantum error correction are used. Instead QKD uses post-processing, and QSDC uses simply classical error correction [1,2]. Therefore, the part about this in QKD in the introduction should be modified. In addition, one related reference is suggested [3].
[1] Kwek, Leong-Chuan, et al. "Chip-based quantum key distribution." AAPPS Bulletin 31 (2021): 15
[2] Zhang, Haoran, et al. "Realization of quantum secure direct communication over 100 km fiber with time-bin and phase quantum states." Light: Science & Applications 11.1 (2022): 83.
[3] Li Ding, Haowen Wang, Yinuo Wang, Shumei Wang, "Based on Quantum Topological Stabilizer Color Code Morphism Neural Network Decoder", Quantum Engineering, vol. 2022, Article ID 9638108, 8 pages, 2022.
The English is fairly good.
Author Response
Dear Reviewer,
Thank you for your comments concerning our manuscript entitled “ Quantum LDPC Codes Based on Cocyclic Block Matrices” (ID: entropy-2558116). Those comments are valuable and very helpful. We have read through comments carefully and have made corrections. We have submitted a list of responses to reviewer comments and the revised version, expecting your kind consideration.
Q: It should be pointed that QEEC is of only nomial siginificance at present because in both QKD and QSDC, no quantum error correction are used. Instead QKD uses post-processing, and QSDC uses simply classical error correction [1,2]. Therefore, the part about this in QKD in the introduction should be modified. In addition, one related reference is suggested [3].
Response: According to your helpful comments, we have modified the introduction in our revised version. Furthermore, we also cited the valuable references.
We would love to thank you for allowing us to resubmit a revised copy of the manuscript and highly appreciate your time and consideration.
With best wishes
Li Yuan

Reviewer 2 Report
Comments on Entropy 2558116 \textbf{``Quantum LDPC Codes Based on Cocyclic Block Matrices"
In this paper, the authors propose a construction method of quantum low density
parity check (LDPC) codes based on cocyclic block matrices. The constructed stabilizer quantum codes can
be obtained from the different rows of the yielded circulant
permutation matrices. The authors use a recursive relation
of block matrix to produce
the generator matrix. As a result, quantum cocyclic codes with long length
are constructed. At last, the performance of quantum LDPC codes is evaluated and compared.
The paper presents a method to construct quantum error-correcting codes with long length. The techniques are novel and the results are interesting. I think the paper is suitable for publication in Entropy after considering the following suggestions and comments.
\begin{enumerate}
\item The length of the considered codes only takes the form $N=4p^{m}$. Can the code length be extended to others in your method?
\item It is well known that the minimum distance is an important parameter of measuring the error-correcting capacity of quantum codes. How the minimum distance of the obtained quantum LDPC codes is? It should give an evaluation or description about it.
\item The authors mention the resulting quantum LDPC codes are no 4-cycles in the Tanner graph. However, it seems that a complete proof has not been provided. Please explain or prove this in a proper place.
\item On presentation, there exist some typos and syntax errors, and it is poorly editing. It needs to make revisions carefully. We list a small part of them here. In Line 15 of Introduction, revise ``[1],[2],[3],[4],[5],[6],[7]" as ``[1]-[7]". In Line 4 of the left column of Page 2, revise ``$\sigma_{z}$" as ``$\sigma_{y}$". In Line 15 of the left column of Page 2, revise ``For" as ``for". In Line 21 of the right column of Page 2, which reference does the question mark [?] stand for?
\end{enumerate}

average
Author Response
Dear Reviewer,
Thank you for your comments concerning our manuscript entitled “ Quantum LDPC Codes Based on Cocyclic Block Matrices” (ID: entropy-2558116). Those comments are valuable and very helpful. We have read through comments carefully and have made corrections. We have submitted a list of responses to reviewer comments and the revised version, expecting your kind consideration.
Q1. The length of the considered codes only takes the form $N=4p^{m}$. Can the code length be extended to others in your method?
Response1:Because the constructed matrices are based on a base matrix of 4 times 4, the code lengths in our paper are only $N=4^m$ and $N=4p^m$ respectively. In the future, we will delve deeper into the construction methods of quantum codes of other lengths
Q2. It is well known that the minimum distance is an important parameter of measuring the error-correcting capacity of quantum codes. How the minimum distance of the obtained quantum LDPC codes is? It should give an evaluation or description about it.
Response2:According to your helpful suggestions, we give an evaluation about the minimum distance. In fact, the obtained generator matrix belongs to the regular matrix, hence at least its minimum distances are bounded above by $O(N^{1/2} \log N)$ as the code length $N \rightarrow \infty $.
Q3. The authors mention the resulting quantum LDPC codes are no 4-cycles in the Tanner graph. However, it seems that a complete proof has not been provided. Please explain or prove this in a proper place.
Response3: We further explain the fact that there is no 4-cycles in the Tanner graph in Section III C (p. 6).
Q4. On presentation, there exist some typos and syntax errors, and it is poorly editing. It needs to make revisions carefully. We list a small part of them here. In Line 15 of Introduction, revise ``[1],[2],[3],[4],[5],[6],[7]" as ``[1]-[7]". In Line 4 of the left column of Page 2, revise ``$\sigma_{z}$" as ``$\sigma_{y}$". In Line 15 of the left column of Page 2, revise ``For" as ``for". In Line 21 of the right column of Page 2, which reference does the question mark [?] stand for?
Response4: In the initial manuscript, we adopted the IEEE Latex template. This format ``[1],[2],[3],[4],[5],[6],[7]" in Introduction will be finally revised if the paper is ultimately accepted. Furthermore, some typos and syntax errors in manuscript have been revised.
We would love to thank you for allowing us to resubmit a revised copy of the manuscript and highly appreciate your time and consideration.
With best wishes
Li Yuan

Reviewer 3 Report
I am submitting my assessment of the manuscript. This manuscript falls short of reaching the level required for publication. Having reviewed only Sections 1 and 2, I found the content quite frustrating. Consequently, I cannot proceed to assess Sections 3 and beyond. I strongly recommend rejection at this point. However, once the substantial issues are resolved, resubmission could be considered.
I have listed technical points of concern below:
-
ABSTRACT: The phrasing "The obtained stabilizer quantum codes (SQCs) can be obtained" is awkward.
-
INTRODUCTION, LEFT COLUMN: I advise against using "kernel" in the sentence "how to design ... is the kernel," as "kernel" has a different meaning in coding theory.
-
INTRODUCTION, RIGHT COLUMN: References [9, 10, 11, 12, 13] appear to be related to classical LDPC codes, while [14, 15, 16, 17, 18] pertain to quantum codes. However, [12] and [13] discuss quantum LDPC codes, so they should be moved to the latter category.
-
INTRODUCTION, RIGHT COLUMN: The paper's strength lies in its absence of "4-cycles" in Tanner graphs. Not citing the paper that first achieved this for quantum LDPC codes is unusual. This paper is known for achieving quantum Quasi-Cyclic LDPC codes using algebraic methods.
HAGIWARA, Manabu; IMAI, Hideki. Quantum quasi-cyclic LDPC codes. In: 2007 IEEE International Symposium on Information Theory. IEEE, 2007. p. 806-810.
-
Section 2-A, The 4th Line: Is $\sigma_x = \sigma_x \sigma z$ correct?
-
Section 2-A, The 4th Line: $\mathcal{P}$ does not form a group since it is not closed under multiplication. Please verify the definition of a group.
-
Section 2-A, The 7th Line: "code words" should be "codewords."
-
Section 2-A, The 9th Line: What is "N-qubit group"?
-
Section 2-A, The 4th Line after (1): "a Hilbert codewords space" should be "a Hilbert codeword space."
-
Section 2-A, (4): Too many typos.
-
Section 2-A, (5): "i_n" should be "i_N."
-
Section 2-A, (6): Do not use "$\cdot" in a different manner on the LHS and RHS—it is inconsistent.
-
Section 2-B: $x_n$ should be $x_N$, and $z_n$ should be $z_N$.
-
Section 2-B: Please carefully review [?].
See the comments and suggestions.
Author Response
Dear Reviewer,
Thank you for your comments concerning our manuscript entitled “ Quantum LDPC Codes Based on Cocyclic Block Matrices” (ID: entropy-2558116). Those comments are valuable and very helpful. We have read through comments carefully and have made corrections.
Q1. INTRODUCTION, RIGHT COLUMN: References [12, 13, 14, 15] appear to be related to classical LDPC codes, while [16, 17, 18, 19, 20, 21, 22, 23] pertain to quantum codes. However, [16] and [17] discuss quantum LDPC codes, so they should be moved to the latter category.
Response 1:We have adjusted the citation number of the reference, and also added some other references.
Q2. Having reviewed only Sections 1 and 2, I found the content quite frustrating. Consequently, I cannot proceed to assess Sections 3 and beyond.
Response 2:I'm sorry for all the errors in the initial draft. Thank you very much for your helpful suggestions. We have corrected the error you raised and further checked the content of the latter sections. Furthermore, we also cited the valuable reference about quantum Quasi-Cyclic LDPC codes using algebraic methods.
We would love to thank you for allowing us to resubmit a revised copy of the manuscript and highly appreciate your time and consideration.
With best wishes
Li Yuan

Round 2
Reviewer 2 Report
The authors respond all suggestions in my early report. My several concerns have been considered in the revised version. In particular, the minimum distance of the obtained quantum LDPC codes is estimated. The presentation of the paper has been greatly improved. I approve the work of the manuscript this time. In addition, I don't find a mathematical problem in the arguments. In short, I recommend acceptance of the revised manuscript.
The paper is written well and the quality of English language is good.
Author Response
We sincerely thank you for your valuable revision suggestions on our manuscript. Your affirmation will greatly encourage our in-depth research in the field of quantum information.
Reviewer 3 Report
I have briefly reviewed the manuscript and noticed that one of the main results is inaccurately stated. I recommend a careful revision of the manuscript. Specifically, I suggest that the authors address the following points:
- Provide a clear definition of "girth/cycle" for "stabilizer code."
- Explain if there is the difference in "girth/cycle" between "stabilizer code" and "CSS code."
- Clarify whether the "no 4-cycle" claim in this manuscript is the same as in [18].
- Explicitly present the proof of achieving "no 4-cycle" using the method proposed in this manuscript, labeling it as a "Theorem" and providing a detailed "Proof."
Additionally, I noted that references have been added to the last page. However, the style of referencing needs improvement. Please carefully review the references for accuracy. For instance:
- Include a comma "," after the author's name in [1].
- Ensure consistent formatting for [14] and [15], as they are published in the same journal.
- Correct the author names in [18]. The family name is "Hagiwara" and the given name is "Manabu." The family name is "Imai" and the given name is "Hideki."
I strongly recommend addressing these points thoroughly in your revision.
It is poor.
Author Response
Dear Reviewer,
I am very grateful to your valuable comments for our manuscript. Your meticulous and responsible works have greatly contributed to the improvement of the manuscript. Your questions were answered below, expecting your kind consideration.
Q1: Provide a clear definition of "girth/cycle" for "stabilizer code."
Response 1: We have defined the girth and cycle of quantum LDPC codes respectively in revised version.
Q2: Explain if there is the difference in "girth/cycle" between "stabilizer code" and "CSS code."
Response 2: There is no difference in "girth/cycle" between "stabilizer code" and "CSS code", as their check matrices are composed of two classical matrices and both detect two types of errors (X-error and Z-error).
Q3: Clarify whether the "no 4-cycle" claim in this manuscript is the same as in [18].
Response 3: We have clarified that claim in this manuscript is different from [18].
Q4: Explicitly present the proof of achieving "no 4-cycle" using the method proposed in this manuscript, labeling it as a "Theorem" and providing a detailed "Proof."
Response 4: We have implemented the helpful suggestions you have put forward.
Q5: Additionally, I noted that references have been added to the last page. However, the style of referencing needs improvement. Please carefully review the references for accuracy. For instance:
- Include a comma "," after the author's name in [1].
- Ensure consistent formatting for [14] and [15], as they are published in the same journal.
- Correct the author names in [18]. The family name is "Hagiwara" and the given name is "Manabu." The family name is "Imai" and the given name is "Hideki."
Response 5: We have further improved the format of the references. In addition, this format will be uniformly modified according to the journal “Entropy” if the manuscript is accepted.
Thank you again for your dedication and effort to this manuscript.
Kind regards,
Yuan Li

Round 3
Reviewer 3 Report
The author's reply contains contradiction.
The author said "There is no difference in "girth/cycle" between "stabilizer code" and "CSS code." It implies that the answer of Q3 must be yes. Both of [18] and the manuscript claim "the constructed parity check-matrices has no 4-cycle." However, the author answered no. It does not make sense.
The author also ignored to correct the names of authors of [18]. It must be
[18] M. Hagiwara, H. Imai, Quantum quasi-cyclic LDPC codes. In: 2007
IEEE International Symposium on Information Theory, IEEE. 806-810,
(2007).
Need to improve.
Author Response
Dear Reviewer,
I am very grateful to your valuable comments for our manuscript. Your questions were answered below, expecting your kind consideration again.
Q1: The author said "There is no difference in "girth/cycle" between "stabilizer code" and "CSS code." It implies that the answer of Q3 must be yes. Both of [18] and the manuscript claim "the constructed parity check-matrices has no 4-cycle." However, the author answered no. It does not make sense.
Response 1: I'm sorry for misunderstanding the question you raised in Review Report (Round 2) and causing confusion for you. Therefore, I reply again that Q3 in the second round of responses is “We have clarified that claim in this manuscript is not different from [18]. ”
Q2: The author also ignored to correct the names of authors of [18]. It must be [18] M. Hagiwara, H. Imai, Quantum quasi-cyclic LDPC codes. In: 2007 IEEE International Symposium on Information Theory, IEEE. 806-810, (2007).
Response 2: We have corrected the expression of the author's name in this reply version.
Thank you again for your dedication and effort to this manuscript.
Kind regards,
Yuan Li

Round 4
Reviewer 3 Report
The manuscript is properly revised.
Not good quality.